# Temporal-attentive Covariance Pooling Networks for Video Recognition

**Zilin Gao[†], Qilong Wang[‡], Bingbing Zhang[†], Qinghua Hu[‡], Peihua Li[†*]**

[†]School of Information and Communication Engineering, Dalian University of Technology
[‡]College of Intelligence and Computing, Tianjin University
`gzl@mail.dlut.edu.cn, qlwang@tju.edu.cn, icyzhang@mail.dlut.edu.cn`
`huqinghua@tju.edu.cn, peihuali@dlut.edu.cn`

## Abstract

For video recognition task, a global representation summarizing the whole contents of the video snippets plays an important role for the final performance. However, existing video architectures usually generate it by using a simple, global average pooling (GAP) method, which has limited ability to capture complex dynamics of videos. For image recognition task, there exist evidences showing that covariance pooling has stronger representation ability than GAP. Unfortunately, such plain covariance pooling used in image recognition is an orderless representative, which cannot model spatio-temporal structure inherent in videos. Therefore, this paper proposes a Temporal-attentive Covariance Pooling (TCP), inserted at the end of deep architectures, to produce powerful video representations. Specifically, our TCP first develops a temporal attention module to adaptively calibrate spatio-temporal features for the succeeding covariance pooling, approximatively producing attentive covariance representations. Then, a temporal covariance pooling performs temporal pooling of the attentive covariance representations to characterize both intra-frame correlations and inter-frame cross-correlations of the calibrated features. As such, the proposed TCP can capture complex temporal dynamics. Finally, a fast matrix power normalization is introduced to exploit geometry of covariance representations. Note that our TCP is model-agnostic and can be flexibly integrated into any video architectures, resulting in TCPNet for effective video recognition. The extensive experiments on six benchmarks (e.g., Kinetics, Something-Something V1 and Charades) using various video architectures show our TCPNet is clearly superior to its counterparts, while having strong generalization ability. *The source code is publicly available.*

## 1 Introduction

Video recognition aims to automatically analyze the contents of videos (e.g., events and actions), and has a wide range of applications, including intelligent surveillance, multimedia retrieval and recommendation. The great progress of deep learning, especially convolutional neural networks (CNNs) [48, 52, 22, 61], remarkably improves performance of video recognition. Since videos in the wild involve complex dynamics caused by appearance changes and variation of visual tempo, powerful video representations potentially provide high performance of video recognition. However, most of the existing video recognition architectures based on 3D CNNs [53, 43, 2, 11, 54, 10] or 2D CNNs [35, 27, 33, 39] usually generate the final video representations by leveraging a global average pooling (GAP). Such GAP simply computes the first-order statistics of convolution features in a temporal orderless manner, which discards richer statistical information inherent in spatio-temporal features and has limited ability to capture complex dynamics of videos.

---

[*]Corresponding author.

35th Conference on Neural Information Processing Systems (NeurIPS 2021).

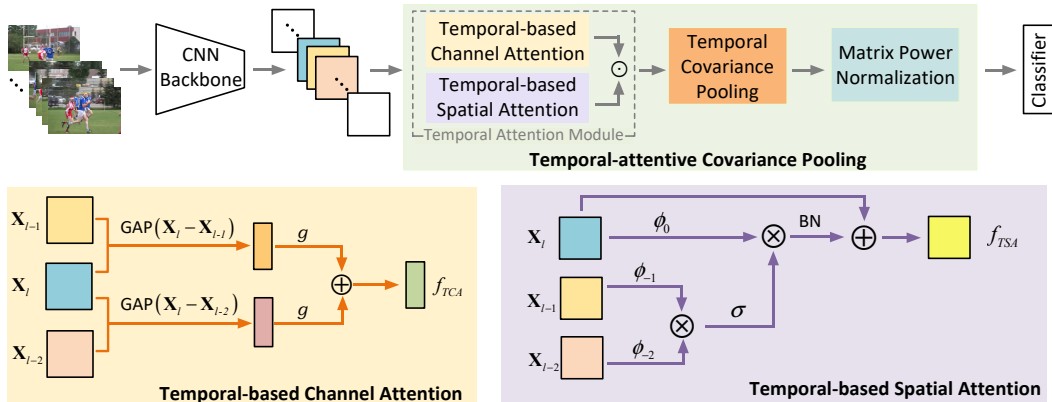

Figure 1: Overview of our TCPNet for video recognition. Its core is a Temporal-attentive Covariance Pooling (TCP) integrated at the end of deep recognition architectures to produce powerful video representations. Specifically, our TCP consists of a temporal attention module for adaptively calibrating spatio-temporal features, a temporal covariance pooling to characterize intra-frame correlations and inter-frame cross-correlations of the calibrated features in a temporal manner, and a fast matrix power normalization for use of geometry of covariances. Refer to Section 3.2 for details.

There are a few researches studying global high-order pooling for improving performance of video recognition [45, 29, 7]. Sanin et al. [45] propose spatio-temporal covariance descriptors as video representations, where covariance descriptors are built with hand-crafted features of candidate regions and a subset of the optimal descriptors are learned by LogitBoost classifiers. However, such a method is designed based on the idea of the classical shallow architectures, without making the most of the merits of deep learning paradigm. In [29], two tensor-based feature representations (namely SCK and DCK) are introduced for action recognition, where multiple subsequences and CNN classifier scores are used to compute tensor representations while a support vector machine is trained for classification (see Figures 2&4 and the third paragraph of Section 5.2). Under end-to-end deep architectures, Diba et al. [7] propose a temporal linear encoding (TLE) method for video classification. Specifically, TLE adopts a bilinear pooling [37] or compact bilinear pooling [15] to aggregate the features output by 2D and 3D CNNs as the entire video representations. TLE shows better performance than the original fully-connected (FC) layer, but it fails to fully exploit temporal information and geometry of covariance representations, limiting effectiveness of covariance pooling for video recognition. Recently, there also exist evidences showing that global covariance (second-order) pooling can significantly improve performance of deep CNNs in various *image-related* tasks, such as fine-grained visual recognition [37, 15], large-scale image classification [31, 30, 60] and visual question answering [13, 68]. However, these methods aim to generate image representations, without considering important temporal information inherent in videos. Meanwhile, naive usage of *image-related* covariance pooling (called *plain covariance pooling* in this work) technique for *video-related* tasks cannot make full use of the advantage of covariance pooling [63].

As discussed previously, though global covariance pooling shows the potential to improve recognition performance, aforementioned works suffer from the following drawbacks: (1) Shallow approach [45] fails to make the most of the merits of deep learning paradigm; (2) Plain covariance pooling methods [37, 15, 31, 30, 60] as well as TLE [7] cannot adequately model complex temporal dynamics, and in the meanwhile TLE considers no geometry of covariances. To overcome these issues, this paper proposes a Temporal-attentive Covariance Pooling (TCP), which can be integrated into existing deep architectures to produce powerful video representations. Different from the plain covariance pooling methods which perform orderless aggregation along temporal dimension, our TCP presents a temporal pooling of attentive covariance representations to capture complex temporal dynamics. Specifically, as illustrated in Figure 1, our TCP consists of a temporal attention module, a temporal covariance pooling, and a fast matrix power normalization of covariances. The temporal attention module is developed to adaptively calibrate spatio-temporal features for the subsequent covariance pooling. The covariance pooling of the calibrated features can be approximatively regarded as producing attentive covariance representations, having the ability to handle complex dynamics. Then, temporal pooling of the attentive covariance representations (i.e., temporal covariance pooling) characterizes both intra-frame correlations and inter-frame cross-correlations of the calibrated features, which can fully harness the complex temporal information. The fast matrix power normalization [30] is finally

introduced to exploit geometry of covariance spaces. Note that our TCP is model-agnostic and can be flexibly integrated into existing 2D or 3D CNNs in a plug-and-play manner, resulting in an effective video recognition architecture (namely TCPNet).

The overview of our TCPNet is shown in Figure 1. To verify its effectiveness, extensive experiments are conducted on six video benchmarks (i.e., Mini-Kinetics-200 [66], Kinetics-400 [2], Something-Something V1 [19], Charades [46], UCF101 [49] and HMDB51 [23]) using various deep architectures (e.g., TSN [59], X3D [10] and TEA [33]). The contributions of our work are summarized as follows. (1) We propose an effective TCPNet for video recognition, where a model-agnostic Temporal-attentive Covariance Pooling (TCP) integrated at the end of existing deep architectures can produce powerful video representations. (2) To our best knowledge, our TCP makes the first attempt to characterize both intra-frame and inter-frame correlations of the calibrated spatio-temporal features in a temporal manner, which approximatively performs temporal pooling of the attentive covariance representations. The proposed TCP provides a compelling alternative to GAP for improving recognition performance of deep video architectures. (3) We perform a comprehensive study of global covariance pooling for deep video recognition architectures, and conduct extensive experiments on six challenging video benchmarks. The results show our TCPNet has strong generalization ability while performing favorably against state-of-the-arts.

## 2 Related Works

### 2.1 Deep Architectures for Video Recognition

Video data usually consists of appearance information and temporal motion. Two-stream architectures [47, 12] are specially developed for separately capturing the two kind of information. They obtain competitive performance, but suffer from huge computational cost due to offline computation of optical flow. Extending 2D convolutions to 3D ones, 3D CNNs achieve state-of-the-art performance by taking only RGB inputs [53, 2, 43, 66, 54, 11, 10], which can simultaneously learn appearance and temporal features. Furthermore, a family of attention models based on idea of non-local [61, 64, 3, 21] are proposed to further improve performance of 3D CNNs by adaptively calibrating spatio-temporal features. Recently, integration of temporal modeling with 2D CNNs achieves promising results in a more efficient way [35, 27, 9, 50, 33, 39, 40], where 2D convolutions extract appearance features while temporal modules (e.g., channel shift, temporal difference, and temporal convolution) are used to capture motion information. However, these video recognition architectures usually generate the final video representations by leveraging a global average pooling (GAP), which has the limited ability to capture complex dynamics of videos. Some works exploit classical encoding methods (i.e., VLAD [26] and Fisher vector [44]) to replace GAP for aggregating spatio-temporal features [18, 57]. Different from aforementioned methods, we propose a temporal-attentive covariance pooling (TCP) to replace GAP for producing powerful video representations in deep architectures.

### 2.2 Deep Covariance (Second-order) Pooling

Global covariance (second-order) pooling has been successfully applied to various image-related tasks, such as image classification [37, 25, 15, 31, 30, 16, 60] and visual question answering [13, 68]. These works show global covariance pooling is much better than the original GAP, and appropriate use of geometry of covariances plays a key role for achieving high performance [25, 31]. However, these works have no ability to capture temporal information inherent in video data, and naive usage of them cannot fully exploit advantages of covariance pooling [63]. For modeling videos based on deep second-order pooling, TLE [7] makes an attempt to use bilinear pooling [37] and CBP [15] for aggregating spatio-temporal features in an end-to-end manner. However, this method makes no full use of temporal information and geometry of covariance representations. Besides, some researches [17, 71, 34] integrate local second-order statistics into deep architectures for improving performance of video recognition. Specifically, AP [17] introduces second-order statistics by designing an element-wise multiplication of multiple-branch based on rank-1 approximation. Sharing similar idea, ABM [71] and TBN [34] apply element-wise multiplication to various frames for capturing motion information. Different from above works, our proposed TCP performs a global temporal pooling of attentive covariance representations, which can handle complex temporal dynamics and exploit geometry of covariances. Besides, our TCPNet has the ability to fully harness the merits of deep learning paradigm, compared to shallow approach [45].

# 3 Proposed Method

In this section, we first briefly introduce the plain covariance pooling for modeling videos. Then, we describe in detail our proposed temporal-attentive covariance pooling (TCP), which consists of a temporal attention module, a temporal covariance pooling, and a fast matrix power normalization. Finally, we instantiate our TCP with integration of existing deep video architectures, resulting in temporal-attentive covariance pooling networks (TCPNet).

## 3.1 Plain Covariance Pooling for Modeling Videos

For the existing video architectures, global average pooling (GAP) is widely used to generate the final video representations. Suppose we have a total of $L$ feature tensors output by some deep architecture. Let $H, W$ and $C$ be spatial height, width and channel number of one feature tensor. We denote by $\mathbf{X}_l \in \mathbb{R}^{N \times C}$ the matrix obtained from the $l$-th feature tensor where $N = H \times W$ and each row $\mathbf{x}_{l,n}$ is a $C$-dimensional feature. GAP computes the final video representation $\mathbf{p}_{GAP} \in \mathbb{R}^{1 \times C}$ as

$$\mathbf{p}_{GAP} = \frac{1}{LN} \sum_{l=1}^{L} \sum_{n=1}^{N} \mathbf{x}_{l,n}, \tag{1}$$

where $L$ indicates temporal resolution. As shown in Eqn. (1), from statistical view, GAP estimates first-order moment while higher-order statistics containing richer information are discarded.

Compared to GAP, covariance pooling can capture richer statistics by modeling correlation among the features. For video data, plain covariance pooling generates video representation $\mathbf{P}_{PCP} \in \mathbb{R}^{C \times C}$ by aggregating the features as

$$\mathbf{P}_{PCP} = \frac{1}{LN} \sum_{l=1}^{L} \sum_{n=1}^{N} \mathbf{x}_{l,n}^T \mathbf{x}_{l,n} = \frac{1}{L} \sum_{l=1}^{L} \frac{1}{N} \mathbf{X}_l^T \mathbf{X}_l, \tag{2}$$

where $T$ indicates the matrix transpose. Although $\mathbf{P}_{PCP}$ in Eqn. (2) has better representation ability than $\mathbf{p}_{GAP}$, both of them perform orderless aggregation along temporal dimension based on simple average operation, lacking the ability to handle complex temporal dynamics. In addition, naive use of $\mathbf{P}_{PCP}$ only obtains small gains (see Section 4) as geometry of covariance spaces is neglected.

## 3.2 Temporal-attentive Covariance Pooling (TCP)

To overcome the drawbacks of plain covariance pooling, we propose a temporal-attentive covariance pooling (TCP) to capture richer statistics of spatio-temporal features and handle complex dynamics.

### 3.2.1 Temporal Covariance Pooling

We regard the video signal as a stationary stochastic process [38] and propose to model its statistics by the second-order moments in a learnable manner. Specifically, we perform a temporal-attentive covariance pooling $\mathbf{P}_{TCP}$ for summarizing the statistics of the videos:

$$\mathbf{P}_{TCP} = \text{TP}_L(\{\mathbf{C}_1, \ldots, \mathbf{C}_l, \ldots, \mathbf{C}_L\}), \quad \mathbf{C}_l = \frac{1}{N} \mathbf{X}_l^T \mathbf{X}_l, \tag{3}$$

where $\text{TP}_L(\cdot)$ indicates a temporal pooling of frame-wise covariance representations $\{\mathbf{C}_l\}_{l=1,\ldots,L}$. In this work, we achieve $\text{TP}_L(\cdot)$ with a temporal convolution of covariances followed by average pooling:

$$\text{TP}_L(\{\mathbf{C}_1, \ldots, \mathbf{C}_l, \ldots, \mathbf{C}_L\}) = \frac{1}{L} \sum_{l=1}^{L} \text{TC-COV}_\kappa(\mathbf{X}_l), \tag{4}$$

where $\text{TC-COV}_\kappa(\mathbf{X}_l)$ performs temporal convolution of covariance $\mathbf{X}_l$ with kernel size of $\kappa$. By taking $\kappa = 3$ as an example, we achieve $\text{TC-COV}_{\kappa=3}(\mathbf{X}_l)$ by considering the temporal correlations of intra- and inter- frames in a sliding window manner. Thus, $\text{TC-COV}_{\kappa=3}(\mathbf{X}_l)$ is computed as

$$\text{TC-COV}_{\kappa=3}(\mathbf{X}_l) = \underbrace{\mathbf{W}_{-1}^T \mathbf{X}_{l-1}^T \mathbf{X}_{l-1} \mathbf{W}_{-1} + \mathbf{W}_0^T \mathbf{X}_l^T \mathbf{X}_l \mathbf{W}_0 + \mathbf{W}_1^T \mathbf{X}_{l+1}^T \mathbf{X}_{l+1} \mathbf{W}_1}_{\text{intra-frame covariance pooling}} \tag{5}$$

$$+ \underbrace{\mathbf{W}_{-1}^T \mathbf{X}_{l-1}^T \mathbf{X}_l \mathbf{W}_0 + \cdots + \mathbf{W}_0^T \mathbf{X}_l^T \mathbf{X}_{l-1} \mathbf{W}_{-1} + \cdots + \mathbf{W}_1^T \mathbf{X}_{l+1}^T \mathbf{X}_l \mathbf{W}_0}_{\text{inter-frame cross-covariance pooling}},$$

where $\{\mathbf{W}_*\}_{*\in\{-1,0,1\}}$ are a set of learnable parameters. For Eqn. (5), the first part of terms is a combination of frame-wise covariances learned based on geometry-aware projections [20], aiming to characterize intra-frame correlations. Meanwhile, the second part of terms is a combination for learnable cross-covariances between all pairs of different frames, aiming to capture inter-frame correlations. In this work, we model the distributions of stochastic process describing the video signals by second-order moments. Along this line, we can further exploit higher-order (e.g., third-order) moments [5] in a learnable manner for more accurate distribution modeling, which is left for future work.

### 3.2.2 Temporal Pooling of Attentive Covariances

Although Eqn. (5) performs a temporal convolution on covariances, both intra-frame covariances (e.g., $\mathbf{C}_{l,l} = \mathbf{X}_l^T\mathbf{X}_l$, $1/N$ is omitted for simplicity) and inter-frame cross-covariances (e.g., $\mathbf{C}_{l,l-1} = \mathbf{X}_l^T\mathbf{X}_{l-1}$) are computed in a specific manner, having limited ability to handle complex temporal dynamics. Therefore, we can calculate attentive covariances (or cross-covariances) to adaptively calibrate importance of covariance representations to account for dynamic changes. However, element-wise attention for covariances (e.g., $s \cdot \mathbf{C}_{l,l}$ or $\mathbf{S} \odot \mathbf{C}_{l,l}$) will break up their geometry. In this work, we approximatively compute attentive covariances by developing a temporal attention module to calibrate the features before temporal covariance pooling:

$$\text{TC-COV}_{\kappa=3}(\widehat{\mathbf{C}}_l) = \underbrace{\cdots + \mathbf{W}_0^T\widehat{\mathbf{C}}_{l,l}\mathbf{W}_0 + \cdots}_{\text{intra-frame covariance pooling}} + \underbrace{\cdots + \mathbf{W}_0^T\widehat{\mathbf{C}}_{l,l+1}\mathbf{W}_1\cdots}_{\text{inter-frame cross-covariance pooling}}, \qquad (6)$$

where $\widehat{\mathbf{C}}_{l,l} = \widehat{\mathbf{X}}_l^T\widehat{\mathbf{X}}_l$, and the features $\widehat{\mathbf{X}}_l$ can be adaptively calibrated by a temporal-based spatial attention $f_{TSA}$ and a temporal-based channel attention $f_{TCA}$, i.e.,

$$\widehat{\mathbf{X}}_l = \left(\mathbf{X}_l \oplus f_{TSA}\left(\mathbf{X}_{l-2}, \mathbf{X}_{l-1}, \mathbf{X}_l\right)\right) \odot f_{TCA}\left(\mathbf{X}_{l-2}, \mathbf{X}_{l-1}, \mathbf{X}_l\right), \qquad (7)$$

where $\oplus$ and $\odot$ represent element-wise addition and element-wise multiplication, respectively. Both $f_{TSA}$ and $f_{TCA}$ are functions of $\mathbf{X}_l$ and its two preceding frames (i.e., $\mathbf{X}_{l-2}$ and $\mathbf{X}_{l-1}$). Next, we will describe how to compute $f_{TSA}$ and $f_{TCA}$.

**Computation of $f_{TSA}$** Given $\mathbf{X}_{l-2}$, $\mathbf{X}_{l-1}$ and $\mathbf{X}_l$, we compute $f_{TSA}$ as

$$f_{TSA}\left(\mathbf{X}_{l-2}, \mathbf{X}_{l-1}, \mathbf{X}_l\right) = BN\left(\sigma[\phi_{-1}(\mathbf{X}_{l-1})\phi_{-2}^T(\mathbf{X}_{l-2})] \otimes \phi_0(\mathbf{X}_l)\right), \qquad (8)$$

where $BN$ means batch normalization while $\sigma$ is softmax operation. $\otimes$ indicates matrix multiplication. $\phi_0$, $\phi_{-1}$ and $\phi_{-2}$ are linear transformations implemented by convolutions. As shown in Eqn. (8), we extract spatial importance of individual features of $\mathbf{X}_l$ by considering relationship among three consecutive frames based on the idea of attention [55], where $\mathbf{X}_{l-1}$, $\mathbf{X}_{l-2}$ and $\mathbf{X}_l$ indicates queries, keys and values, respectively. In this way, we model long-range dependencies of spatial features while attending to the temporal relations based on the inherent smoothness between adjacent frames.

**Computation of $f_{TCA}$** We calibrate the feature channels of $l$-th frame by considering temporal difference. Specifically, we formulate $f_{TCA}$ as

$$f_{TCA}\left(\mathbf{X}_{l-2}, \mathbf{X}_{l-1}, \mathbf{X}_l\right) = \frac{1}{2}g\left(\text{GAP}(\mathbf{X}_l - \mathbf{X}_{l-1})\right) + \frac{1}{2}g\left(\text{GAP}(\mathbf{X}_l - \mathbf{X}_{l-2})\right), \qquad (9)$$

where GAP indicates global average pooling, and $g$ is achieved by two FC layers followed by a Sigmoid function.

After some arrangement, our TCP in Eqn. (6) can be efficiently implemented by a temporal convolution of the calibrated features with kernel size of $1 \times 1 \times \kappa$ followed by a covariance pooling:

$$\mathbf{P}_{TCP} = \frac{1}{L}\sum_{l=1}^{L}\left[\text{TC}_{1\times1\times\kappa}(\widehat{\mathbf{X}}_l)\right]^T\left[\text{TC}_{1\times1\times\kappa}(\widehat{\mathbf{X}}_l)\right]. \qquad (10)$$

where $\text{TC}_{1\times1\times\kappa}(\widehat{\mathbf{X}}_l)$ performs temporal convolution on $\widehat{\mathbf{X}}_l$ with kernel size of $1 \times 1 \times \kappa$. Finally, let us compare GAP, plain covariance pooling and our TCP from the view of stochastic process. Since the video signals are high-dimensional, spanning temporal and spatial dimensions, their distribution is very complex. GAP (resp. plain covariance pooling) only considers mean and variances of *one single time*, while our TCP models two different times, capturing temporal dynamics. In addition, as the features possess spatial structure, various features (either nearby or long-range features) may function differently in the interplay of different times, but GAP and plain covariance pooling neglect this interplay. In contrast, our TCP can attend to this by using idea of attention.

### 3.2.3 Fast Matrix Power Normalization

Our TCP produces a covariance representation for each video. It is well known that the space of covariances is a Riemannian manifold [1, 42], which has geometrical structure. The appropriate use of geometry plays a key role for the effectiveness of covariance representations. Under deep architectures, recent works [31, 60] show matrix power normalization approximatively yet effectively utilizes geometry of covariance space, and is superior to its counterparts (e.g., Log-Euclidean metrics [37] and element-wise power normalization [25]). Here we exploit a fast matrix power normalization method [30] for efficiency and effectiveness. Specifically, for each covariance representation $\mathbf{P}_{TCP}$ output by our TCP, we compute its approximate matrix square root[1] as follows:

$$\text{Iteration:} \{\mathbf{Q}_k = \frac{1}{2}\mathbf{Q}_{k-1}(3\mathbf{I} - \mathbf{R}_{k-1}\mathbf{Q}_{k-1}); \mathbf{R}_k = \frac{1}{2}(3\mathbf{I} - \mathbf{R}_{k-1}\mathbf{Q}_{k-1})\mathbf{R}_{k-1}\}_{k=1,\dots,K}, \quad (11)$$

where $\mathbf{Q}_0 = \mathbf{P}_{TCP}$ and $\mathbf{R}_0 = \mathbf{I}$ is the identity matrix. After $K$ iterations, we have $\mathbf{P}_{TCP}^{\frac{1}{2}} \approx \mathbf{Q}_K$. Finally, $\mathbf{P}_{TCP}^{\frac{1}{2}}$ is used as video representation for classification. In this paper, $K$ is set to 3 throughout all the experiments. Note that Eq. (11) can be efficiently implemented on GPU due to involving of only matrix multiplications, and more details can refer to [30].

### 3.3 Temporal-attentive Covariance Pooling Networks (TCPNet)

In this subsection, we instantiate our TCP with existing deep video architectures, which yield powerful TCPNet for video recognition. As shown in Figure 1, given any specific video architecture (e.g., TSN [59] with 2D ResNet-50 [22] or 3D CNNs [10]), our TCPNet inserts the proposed TCP after the last convolution layer, which replaces the original GAP to produce the final video representations for classification. Note that we introduce $1 \times 1 \times 1$ convolutions between the last convolution layer and our TCP, which reduce dimension of convolution features and control size of video representations. Our TCP is lightweight: taking 8256-dimensional TCP (default setting) as an example, parameters and FLOPs of our TCP are 3.3M and 1.2G for 8 frames of $224 \times 224$ inputs, respectively. Compared to backbones of 2D ResNet-50 (24.3M and 33G) and 2D ResNet-152 (59.0M and 91.5G), our TCP brings about extra 13.6% (5.6%) parameters and 3.6% (1.3%) FLOPs, respectively.

## 4 Experiments

To evaluate our proposed method, we conduct experiments on six video benchmarks, including Mini-Kinetics-200 (Mini-K200) [66], Kinetics-400 (K-400) [2], Something-Something V1 (S-S V1) [19], Charades [46], UCF101 [49] and HMDB51 [23]. We first describe implementation details of our method, and then make ablation studies on Mini-K200. Furthermore, we compare with counterparts and state-of-the-arts on K-400, and finally show generalization of our TCPNet on remaining datasets.

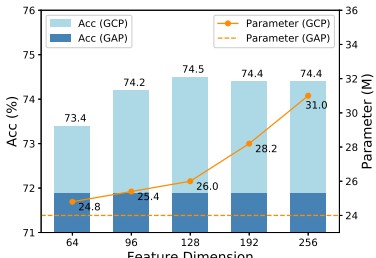

Figure 2: Effect of feature dimension.

| Method | Top-1 (%) | Top-5 (%) |
|---|---|---|
| GAP (Baseline) | 71.9 | 90.6 |
| Plain GCP | 72.1 | 90.8 |
| Plain GCP + MPN | 74.5 | 91.8 |
| $+f_{TCA}$ | 75.1 | 92.2 |
| $+f_{TSA}$ | 75.6 | 92.5 |
| $+\text{TC-COV}_{\kappa=5}$ | **76.1** | **92.7** |

Table 1: Impact of various key components.

**Implementation Details.** To construct our TCPNet, we integrate the proposed TCP with several video architectures, including TSN [59] with 2D CNNs (i.e., ResNet-50 [22] and ResNet-152 [22]), X3D-M [10] and 2D CNNs with temporal modeling (e.g., TEA [33]). ResNet-50 (R50) and ResNet-152 (R152) are pre-trained on ImageNet-1K and ImageNet-11K, respectively. As suggested in [31, 30], we discard the last downsampling operation when covariance representations are computed. Here we describe the settings of hyper-parameters on Mini-K200 and K-400. For training our TCPNet

---

[1]The matrix square root is a special case of matrix power normalization with the power of 1/2, usually achieving the best performance.

| Method | Dim. | Top-1 (%) | Top-5 (%) |
|---|---|---|---|
| GAP | 2048 | 71.9 | 90.6 |
| BCNN [37] | 16384 | 72.7 | 91.4 |
| CBP [15] | 8256 | 72.5 | 91.5 |
| TLE (BCNN-A) [7] | 8256 | 72.2 | 91.0 |
| TLE (CBP-A) [7] | 8256 | 72.2 | 90.6 |
| iSQRT [30] | 8256 | 74.5 | 91.8 |
| iSQRT-A | 8256 | 72.0 | 90.5 |
| iSQRT-B | 8256 | 74.3 | 91.2 |
| TCP (Ours) | 8256 | **76.1** | **92.7** |

Table 2: Comparison of various global pooling methods on Mini-K200.

| Method | Backbone | Input | Top-1 (%) |
|---|---|---|---|
| TBN [34] | 2D R18 | 112 | 69.5 |
| MARS [4] | 3D RX101 | 112 | 72.8 |
| V4D [70] | 3D R50 | 224 | 80.7 |
| BAT [3] | 2D R50 | 224 | 70.6 |
| RMS [28] | 3D Slow50 | 224 | 78.6 |
| CGNL [69] | 3D R101 | 224 | 79.5 |
| TCPNet (Ours) | 2D R50 | 112 | **78.3** |
| TCPNet (Ours) | 2D R50 | 224 | **80.7** |

Table 3: Comparison of state-of-the-arts on Mini-K200.

with 2D CNNs, we adopt the same data augmentation strategy as [59], and number of segments is set to 8 or 16. A dropout with a rate of 0.5 is used for the last FC layer. TCPNet is optimized by mini-batch stochastic gradient descent (SGD) with a batch size of 96, a momentum of 0.9 and a weight decay of 1e-4. The whole networks are trained within 50 epochs, where initial learning rate is 0.015 and decay by 0.1 every 20 epochs. For training our TCPNet with X3D-M, we process the images followed by [10], and 16 frames are sampled as inputs. The SGD with cosine training strategy is used to optimize the network parameters within 100 epochs, and the initial learning rate is set to 0.1. For training our TCPNet with TEA, we first train the backbone of TEA (i.e. Res2Net50 [14]) with covariance pooling on ImageNet-1K, and then fine-tune TCPNet with TEA using the same settings as [33]. During inference, the strategies of 1 crop $\times$ 1 clip and 3 crops $\times$ 10 clips are used for ablation study and comparison with state-of-the-arts. The training settings for the remaining datasets will be specified in the following parts. All programs are implemented by Pytorch and run on a PC equipped with four NVIDIA Titan RTX GPUs.

## 4.1 Ablation Study on Mini-K200

For efficiency, we make ablation studies on Mini-K200 with input size of $112 \times 112$, and TSN with R50 is used as backbone model. We first assess the effects of feature dimension and key components on our TCP, and then compare with different global pooling methods and state-of-the-arts.

**Effect of Feature Dimension**  As described in section 3.3, we introduce $1 \times 1 \times 1$ convolutions before our TCP to reduce dimension of convolution features, controlling the size of covariance representations. Here we assess the effect of feature dimension on recognition accuracy. Note that we do not use temporal attention module and temporal pooling for eliminating their effects and guaranteeing generality. Specifically, the feature dimension varies from 64 to 256, and the corresponding results are shown in Figure 2, where GAP is compared as a baseline. From Figure 2 we can see that global covariance pooling (GCP) consistently outperforms GAP. With increase of feature dimension, performance saturates at dimension of 128, but after that parameters significantly increase. GCP with dimension of 128 is better than GAP by 2.6%, and brings extra $\sim$2M parameters. For efficiency and performance, we set feature dimension to 128 throughout the following experiments, producing 8256-dimensional covariance representations after matrix triangulation.

**Impact of Key Components**  Our TCP involves several key components, including temporal attention module, temporal covariance pooling TC-COV$_\kappa$ ($\kappa = 5$ and $\kappa = 9$ for 8 and 16 frame inputs) and matrix power normalization. To evaluate their effects, we give the results of our TCP under various configurations. As shown in Table 1, plain covariance pooling obtains little improvement over the original GAP and matrix power normalization (MPN) brings about 2.4% gains. Note that Plain GCP + MPN is a strong baseline established by our work. Although promising performance is achieved by this baseline, the temporal-based channel attention $f_{TCA}$ achieves 0.6% gains over plain covariance pooling with matrix power normalization, while temporal-based spatial attention $f_{TSA}$ further brings 0.5% gains. Furthermore, TC-COV$_{\kappa=5}$ achieves 0.5% improvement by performing temporal covariance pooling. Our TCP finally obtains the best results (76.1%), outperforming GAP and plain covariance pooling by 4.2% and 4.0% respectively. These results show the effectiveness of each key component in TCP.

(a) GAP *vs.* TCP (Ours) under various architectures.

| Method | Pooling | Top-1 | Top-5 |
|---|---|---|---|
| TSN [59]+R50 | GAP | 70.6 | 89.2 |
| | TCP | 75.3 (↑ 4.7) | 91.7 (↑2.5) |
| TSN [59]+R152 | GAP | 75.9 | 92.1 |
| | TCP | 78.3 (↑ 2.4) | 93.7 (↑1.6) |
| X3D-M [10] | GAP | 76.0 | 92.3 |
| | TCP | 77.0 (↑ 1.0) | 92.7(↑0.4) |
| TEA [33]+R50 | GAP | 75.0 | 91.8 |
| | TCP | 76.8 (↑1.8) | 92.9(↑1.1) |

(b) Comparison with various 3D CNNs.

| Method | Backbone | Fr×Views | Top-1 | Top-5 |
|---|---|---|---|---|
| I3D [2] | 3D Inc. | 64×N/A | 71.1 | 89.3 |
| R(2+1)D [54] | 3D R34 | 16×10 | 72.0 | 90.0 |
| Non-local [61] | 3D R101 | 32×30 | 76.0 | 92.1 |
| ABM [71] | 3D Inc. | 150×1 | 72.6 | N/A |
| CorrNet [56] | 3D R50 | 32×10 | 77.2 | N/A |
| SlowFast [11] | 3D R50 | 64×30 | 77.0 | 92.6 |
| X3D [10] | X3D-M | 16×30 | 76.0 | 92.3 |
| TCPNet (Ours) | X3D-M | 16×30 | **77.0** | **92.7** |

(c) Comparison with various 2D CNNs (8 frames).

| Method | Backbone | Fr×Views | Top-1 | Top-5 |
|---|---|---|---|---|
| TSN [35] | 2D R50 | 8×10 | 70.6 | 89.2 |
| TPN [67] | 2D R50 | 8×10 | 73.5 | N/A |
| TEA [33] | 2D R50 | 8×30 | 75.0 | 91.8 |
| BAT [3] | 2D R50 | 8×30 | 75.8 | N/A |
| GTA [21] | 2D R50 | 8×30 | 75.9 | 92.2 |
| SmallBig [32] | 2D R50 | 8×30 | 76.3 | 92.5 |
| TCPNet (Ours) | TEA R50 | 8× 30 | 76.8 | 92.9 |
| TCPNet (Ours) | TSN R152 | 8× 30 | **78.3** | **93.7** |

(d) Comparison with various 2D CNNs (16 frames).

| Method | Backbone | Fr×Views | Top-1 | Top-5 |
|---|---|---|---|---|
| STM [27] | 2D R50 | 16×30 | 73.7 | 91.6 |
| TSM [35] | 2D R50 | 16×10 | 74.7 | N/A |
| TEA [33] | 2D R50 | 16×30 | 76.1 | 92.5 |
| TEINet [39] | 2D R50 | 16×30 | 76.2 | 92.5 |
| TAM [40] | 2D R50 | 16×12 | 76.9 | 92.9 |
| bL-TAM [9] | 2D R50 | 24×9 | 73.5 | 91.2 |
| TCPNet (Ours) | TEA R50 | 16× 30 | 77.2 | 93.1 |
| TCPNet (Ours) | TSN R152 | 16× 30 | **79.3** | **94.0** |

Table 4: Comparison (in %) of state-of-the-arts on K-400 dataset under various configurations.

**Comparison of Various Global Pooling Methods**    To further verify the effectiveness of our TCP, we compare with eight kinds of global pooling methods, including GAP, BCNN [37], CBP [15], iSQRT [30], TLE (BCNN-A & CBP-A) [7], iSQRT-A and iSQRT-B. Specifically, we compute GAP, BCNN, CBP and iSQRT by using all features $\{\mathbf{X}_1, \mathbf{X}_2, \ldots, \mathbf{X}_L\}$ in temporal orderless manner. For BCNN-A, CBP-A and iSQRT-A, we first perform average pooling along temporal dimension to obtain $\widetilde{\mathbf{X}} = \sum_l^L \mathbf{X}_l$ upon which we use BCNN, CBP and iSQRT. For iSQRT-B, we first apply iSQRT to each frame $\mathbf{X}_l$, and then perform average pooling for all covariances. The results of all methods are presented in Table 2, based on which we can draw the following conclusions. First, all GCP methods outperform GAP, showing that GCP can produce more powerful video representations. It can also be seen that iSQRT performs significantly better than BCNN and CBP, clearly suggesting that the advantage of exploiting geometry of covariance spaces. Second, BCNN, CBP and iSQRT achieve better results than TLE and iSQRT-A/iSQRT-B, respectively; this suggests that the simple strategy to exploit temporal information [7] brings little gains. Finally, our TCP achieves the best results, which outperforms GAP and iSQRT by 4.2% and 1.6% in Top-1 accuracy, respectively.

**Comparison with State-of-the-arts**    At the end of this part, we compare our TCPNet with several state-of-the-arts on Mini-K200. The compared results are given in Table 3. When 112×112 images are used as inputs, our TCPNet outperforms TBN [34] and MARS [4] by 8.8% and 5.5%, respectively. When size of input images increases to 224×224, our TCPNet obtains 80.7% in Top-1 accuracy, performing best among all competing methods. Notably, for backbone of 2D R50, TCPNet improves BAT [3] over 10.1%. Finally, TCPNet with 2D backbone is superior to CGNL [69] and RMS [28] while obtaining the same result with V4D [70]; note that all three latter exploit 3D backbone models.

### 4.2   Comparison with State-of-the-arts on Kinetics-400

In this subsection, we compare our method with state-of-the-arts on K-400 under various configurations. As shown in Table 4 (a), our TCP consistently outperforms the original GAP under video architectures of TSN, X3D-M and TEA, which span all types of popular video architectures. Particularly, TCP achieves larger improvement if less temporal information is learned by backbone models. Table 4 (b) compares different 3D CNNs, where our TCPNet achieves

| Method | Backbone | Top-1 (%) |
|---|---|---|
| TSN [59] | 2D R50 | 19.7 |
| TSM [35] | 2D R50 | 45.6 |
| TPN [67] | 2D R50 | 49.0 |
| TSM + NeXtVLAD [36] | Inc.-Res | 41.9 |
| TSM + PPAC [41] | Inc.-Res | 43.7 |
| GSM [51] | Inc.-V3 | 49.0 |
| TSN + TCP (Ours) | 2D R50 | 38.3 (↑18.6) |
| TSM + TCP (Ours) | 2D R50 | 46.9 (↑1.3) |
| TPN + TCP (Ours) | 2D R50 | 49.6 (↑0.6) |

Table 5: Comparison on S-S.

| Method | Backbone | Frames | mAP (%) |
|---|---|---|---|
| Non-local [61] | 3D R101 | 128 | 39.5 |
| SlowFast [65] | 3D R50-NL | 64 | 38.0 |
| LFB [64] | 3D R101-NL | 32 | 42.5 |
| STRG [62] | 3D R101-NL | 32 | 39.7 |
| TimeCeption [24] | 3D R101 | 128 | 41.1 |
| SlowFast [11] | 3D R101-NL | 128 | 42.5 |
| TCPNet (Ours) | 3D R101-NL | 32 | **42.6** |

Table 6: Comparison on Charades.

| Method | Backbone | Fr | UCF101 | HMDB51 |
|---|---|---|---|---|
| TSN [35] | 2D R50 | 8 | 91.7 | 64.7 |
| TLE‡ [7] | 2D Inc. | 15 | 95.6 | 71.1 |
| TBN [34] | 2D R34 | 8 | 93.6 | 69.4 |
| AVLAD‡ [18] | VGG-16 | 25 | 92.7 | 66.9 |
| PPAC‡ [41] | 2D R152 | 20 | 94.9 | 69.8 |
| ECO [72] | Inc.+3D R18 | 8 | 91.7 | 65.6 |
| TSM [35] | 2D R50 | 8 | 95.9 | 73.5 |
| TSM† [35] | 2D R50 | 8 | 95.2 | 72.0 |
| ECO [72] | Inc.+3D R18 | 16 | 92.8 | 68.5 |
| TEA [33] | 2D R50 | 16 | **96.9** | 73.3 |
| STM [27] | 2D R50 | 16 | 96.2 | 72.2 |
| STC [6] | 3D RX101 | 16 | 92.3 | 65.4 |
| ART [58] | 3D R18 | 16 | 94.3 | 70.9 |
| I3D [2] | 3D Inc. | 64 | 95.4 | 74.5 |
| ABM [71] | 3D Inc. | 64 | 95.1 | 72.7 |
| TCPNet (Ours) | TSN R50 | 8 | 95.1 | 72.5 |
| TCPNet (Ours) | TEA R50 | 8 | 96.4 | **76.7** |

Table 7: Results (% in Top-1) on UCF101 and HMDB51. †:Re-implementation. ‡: RGB+Flow. AVLAD: ActionVLAD.

comparable performance with state-of-the-arts. Particularly, our TCPNet shows similar performance with SlowFast and CorrNet, but uses less input frames. As listed in Table 4 (c) and Table 4 (d), our TCPNet based on TEA with input of 8 frames outperforms other methods, when R50 is used. For input of 16 frames, TCPNet based on TEA is superior to all the compared methods by a clear margin. When R152 is employed, TCPNet obtains state-of-the-art results for inputs of both 8 and 16 frames.

## 4.3 Generalization to Other Benchmarks

We further verify the generalization ability of our TCP on S-S V1, Charades, UCF101 and HMDB51. For S-S V1, we integrate our TCP into three video architectures, i.e., TSN [59], TSM [35] and TPN [67]. Specifically, we train our TCPNet based on TSN and TSM following the settings in [35] with 8 frames and test with 1 clip × 1 crop, while TCPNet based on TPN is trained using the same settings as [67]. As shown in Table 5, our TCP brings 18.6%, 1.3% and 0.6% gains for TSN, TSM and TPN, respectively. Based on TSM backone, our TCP with 8 frame inputs is superior to NeXtVLAD and PPAC with 20 frame inputs by 5.0% and 3.2%, respectively. Besides, TCP outperforms GSM with backbone of Inception-V3 by 0.6%. For Charades dataset, we fine-tune TCPNet with backbone of 3D R101-NL [61] pre-trained on K-400 using learning rate of 0.3 within 50 epochs. The results of different methods in terms of mean Average Precision (mAP) are presented in Table 6, where our TCPNet obtains 42.6% in terms of mAP, outperforming all compared methods. On UCF101 and HMDB51, TCPNet is fine-tuned based on TSN and TEA with backbone of 2D R50 with initial learning rate of 1e-3 within 15 epochs. The results in top-1 accuracy are shown in Table 7, from it we can see that our TCPNet clearly outperforms TSN. Besides, TCPNet based on TSN is superior to other encoding methods (i.e., TLE, TBN, ActionVLAD and PPAC) by a large margin. Lastly, our TCPNet with TEA performs competitively on UCF101 and achieves state-of-the-art results on more challenging HMDB51. These results clearly show our TCP has a good generalization ability.

## 5 Conclusion

In this paper, we propose a temporal-attentive covariance pooling for generating powerful video representations, which can be seamlessly integrated with existing deep models and result in an effective video recognition architecture (namely TCPNet). Extensive experimental comparisons using various deep video architectures demonstrate the effectiveness of our TCPNet, and show our TCP is a compelling alternative to GAP in popular deep architectures for effective video recognition. By considering complex temporal dynamics and geometry of covariances, our TCP is also superior to its covariance pooling counterparts by a clear margin. In future, we will study more powerful global pooling methods (e.g., third-order pooling or mixture model) for more effective video signal modeling, and apply our proposed TCP to other deep architectures, e.g., vision Transformers [8].

## Acknowledgements

Our work was supported by National Natural Science Foundation of China (Grant Numbers 61971086, 61806140, 61471082, 61872118, 61925602), Natural Science Foundation of Tianjin City (Grant No. 20JCQNJC1530) and CCF-Baidu Open Fund (No. 2021PP15002000).

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
