# Supplementary Materials for "Temporal-attentive Covariance Pooling Networks for Video Recognition"

**Zilin Gao[†], Qilong Wang[‡], Bingbing Zhang[†], Qinghua Hu[‡], Peihua Li[†*]**

[†]School of Information and Communication Engineering, Dalian University of Technology
[‡]College of Intelligence and Computing, Tianjin University
gzl@mail.dlut.edu.cn, qlwang@tju.edu.cn, icyzhang@mail.dlut.edu.cn
huqinghua@tju.edu.cn, peihuali@dlut.edu.cn

In the supplementary material, we first conduct additional experiments to evaluate our TCP module (refer to Section A&B&C), including effect of kernel size $\kappa$ in TC-COV$_\kappa$, design of temporal attention module, and more comparison of our TCP with plain GCP + MPN. In Section D, we give an introduction about Kinetics-400 (K-400) and Mini-Kinetics-200 (Mini-K200) datasets. Finally, we make a quality analysis of our TCP by visualizing the learned attention maps in Section E.

Particularly, the experiments in Section A and B are conducted on Mini-K200 by using the same settings with those in Section 4.1 of the main manuscript, i.e., 8- or 16-frame input with the size of $112 \times 112$ while TSN R-50 is employed as backbone. The experiments in Section C are conducted on K-400 using the same settings as in Section 4.2 of the main manuscript, i.e., 8-frame video clips with the spatial size of $224 \times 224$ are used as inputs.

## A  Effect of Kernel Size $\kappa$ on TC-COV$_\kappa$

Our temporal covariance pooling TC-COV$_\kappa$ has a parameter, i.e., kernel size $\kappa$. Here we assess effect of kernel size $\kappa$ on TC-COV$_\kappa$. For 8 frames input, kernel size $\kappa$ varies from 1 to 7 and 16 frames input, $\kappa$ varies from 1 to 15. As listed in Table A1 & A2, $\kappa = 5$ and $\kappa = 9$ respectively achieves the best results for 8 frames input and 16 frames input. They obtain 0.5% and 0.7% gains over those with $\kappa = 1$, respectively. The performance will decrease gradually when $\kappa > 5$ and $\kappa > 9$, interaction among too large range will bring side effect on performance, while increasing computation cost.

| $\kappa$ | 1 | 3 | 5 | 7 |
|---|---|---|---|---|
| Top-1 Acc (%) | 75.6 | 76.0 | **76.1** | 76.0 |
| Top-5 Acc (%) | 92.5 | 92.7 | **92.7** | 92.6 |

Table A1: Results (8 frames) of TCPNet with various $\kappa$ on Mini-K200.

| $\kappa$ | 1 | 3 | 5 | 7 | 9 | 11 | 13 | 15 |
|---|---|---|---|---|---|---|---|---|
| Top-1 Acc (%) | 77.0 | 77.0 | 77.4 | 77.5 | **77.7** | 77.3 | 76.9 | 76.8 |
| Top-5 Acc (%) | 93.0 | 93.1 | 93.3 | 93.3 | **93.5** | 93.1 | 93.0 | 93.0 |

Table A2: Results (16 frames) of TCPNet with various $\kappa$ on Mini-K200.

---

[*]Corresponding author.

35th Conference on Neural Information Processing Systems (NeurIPS 2021).

| Channel-att | Spatial-att | TC-COV | Acc (%) |
|:---:|:---:|:---:|:---:|
| TCA | Non-local [1] | ✓ | 75.6 |
| SE [2] | TSA | ✓ | 75.6 |
| TCA | TSA | ✓ | **76.1** |
| Query | Key | Value | Acc (%) |
| $\mathbf{X}_{l-1}$ | $\mathbf{X}_l$ | $\mathbf{X}_l$ | 75.5 |
| $\mathbf{X}_{l-1}$ | $\mathbf{X}_{l-2}$ | $\mathbf{X}_l$ | **76.1** |

Table A3: Evaluation on temporal attention module design on Mini-K200.

| Backbone | GAP | Plain GCP+MPN | TCP |
|:---:|:---:|:---:|:---:|
| TSN+R50 | 70.6 | 73.1 | **75.3** |
| TSN+R152 | 75.9 | 77.2 | **78.3** |
| X3D-M | 76.0 | 76.5 | **77.0** |
| TEA+R50 | 75.0 | 75.8 | **76.8** |

Table A4: Comparison with GAP and plain GCP + MPN on K-400.

## B  Design of Temporal Attention Module

Here, we evaluate the effect of our temporal attention module. To this end, we first replace our TSA and TCA by using Non-local and SE blocks, respectively. According to the upper part of Table A3, we can draw the following conclusions: (1) Our TSA achieves 0.5% gains over Non-local in accuracy, while having smaller computational cost (i.e., 0.16 GFLOPs of our TSA vs. 0.7 GFLOPs of Non-local). (2) Our TCA outperforms SE by 0.5% in accuracy. For our TSA module, three consecutive frames are adopted to model spatial attention by exploiting temporal information. Here, we develop a variant of TSA to consider only two consecutive frames, which yields 75.5% in top-1 accuracy and is lower than TSA (76.1%) exploiting three consecutive frames. The results above verify the effectiveness of our temporal attention module.

## C  Comparison TCP with Plain GCP + matrix power normalization (MPN)

In this work, we develop a strong baseline, i.e., Plain GCP + matrix power normalization (MPN). Here, we compare our TCP with this strong baseline on K-400. The results are shown in Table A4, from which we can see that our TCP performs better than Plain GCP + MPN by 2.2%, 1.1%, 0.5% and 1.0% with deep video architectures of TSN + R50, TSN + R152, X3D-M and TEA + R50, respectively. Above results show our TCP can achieve non-trivial performance gains over this strong baseline (Plain GCP + MPN), demonstrating the effectiveness of our TCP in capturing complex temporal dynamics.

## D  Details of Kinetics-400 and Mini-Kinetics-200

Kinetics-400 [3] is a large-scale dataset containing 400 action categories, in which training set and validation set have 246K and 20K videos, respectively. The dataset is released by providing YouTube links. Because some links are broken, we use the dataset collected by [1], which has 234,643 and 19,761 videos for training and validation in total, respectively.

Mini-Kinetics-200 [4] dataset involving of 200 categories is a subset of Kinetics-400, where 400 and 25 videos are used for training and validation per category, respectively. We use the same categories and videos sampling strategy in [4]. The full dataset contains 80K training videos and 5K validation videos. Since broken links, we collect 77,161 and 4,988 videos for training and validation in total.

# E  Visualization of Attention Maps Learned by TCP

To give a quality analysis of TCP, we visualize results of the learnt temporal attention using some example videos on K-400. The qualitative results are shown in Figure A1, where we can see that the attention module in our TCP can effectively focus on key moving parts (e.g., hands and food in "tasting food" Figure A1(a)) for action recognition, while suppressing the remaining irrelevant regions.

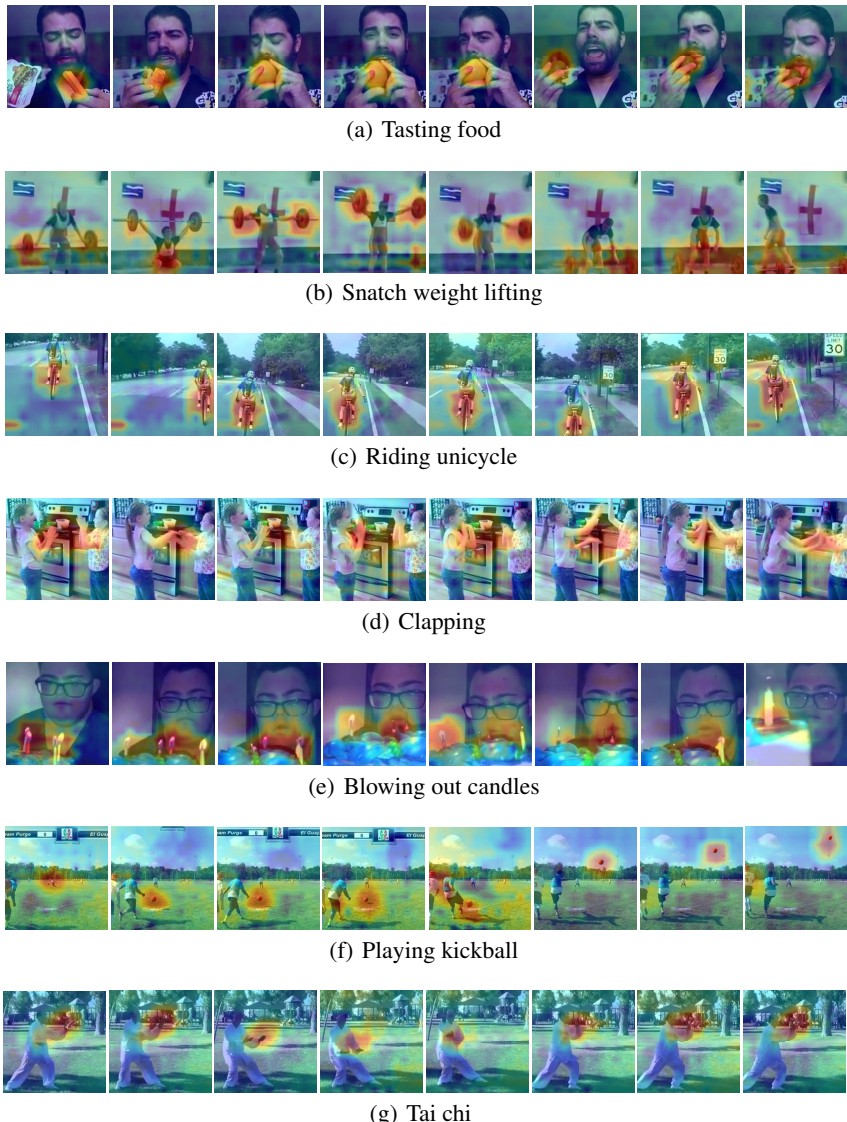

(a) Tasting food

(b) Snatch weight lifting

(c) Riding unicycle

(d) Clapping

(e) Blowing out candles

(f) Playing kickball

(g) Tai chi

Figure A1: Visualization of learned attention maps of some video examples using our TCP on K-400.