# OpenReview forum: "Temporal-attentive Covariance Pooling Networks for Video Recognition"
_NeurIPS.cc/2021/Conference — NeurIPS 2021 Poster_

### Official Review · Reviewer_zp2B · 2021-07-14

**Rating:** 6
**Confidence:** 5

**Summary:**

This paper proposes temporal attentive covariance pooling module for video recognition. This module extracts seconder-order information from video feature tensors to improve the conventional global average pooling layer and it works in a plug-and-play fashion. Temporal-based channel attention and spatial attention are also designed to handle complex temporal dynamics. Extensive results are conducted to show effectiveness of the proposed module.

**Main Review:**

Originality: This paper extends second-order pooling, which is originally developed for image recognition, to the field of video recognition. Instead of directly appling covariance pooling on per-frame basis, this paper proposes temporal-based spatial and channel attention mechanisms to model complex temporal dynamics. Besides, the temporal coviarance pooling scheme takes into acount both intra- and inter-frame covariances. From this point of view, the novelty of this paper is OK.

Quality: Generally, the overall quality of this paper reaches or is slightly above the bar. The motivation of this paper is reasonable  and the performance of the proposed TCP is validated via extensive experiments.

Clarity: It is clear for readers to follow the main idea of this paper.  The draft is well organized and the presentation of this paper is good despite of the following small issues:
1. In eq.5 and eq.6, in my opinion, the first part denotes intra-frame covariance and the second part denotes inter-frame covariance. The terms in the text are also misleading (e.g., line-170, C_{l,l} should be intra-frame covariance, C_{l,l-1} should be inter-frame covariance).
2. There are works which perform local feature integration to generate a compact representation from video feature tensors, such as ActionVLAD [1], NeXtVLAD [2], AttentionClusters[3]. Having some discussions on them or making some comparisons with them is better.

[1] Girdhar, Rohit, et al. "Actionvlad: Learning spatio-temporal aggregation for action classification." Proceedings of the IEEE Conference on Computer Vision and Pattern Recognition. 2017.

[2] Lin, Rongcheng, Jing Xiao, and Jianping Fan. "Nextvlad: An efficient neural network to aggregate frame-level features for large-scale video classification." Proceedings of the European Conference on Computer Vision (ECCV) Workshops. 2018.

[3] Purely Attention Based Local Feature Integration for Video Classification
X Long, G De Melo, D He, F Li, Z Chi, S Wen, C Gan - IEEE TPAMI2020

Significance:  The evaluation results show superiority of TCP, compared to various global pooling methods. The TCP module is also verified to be applicable for various 2D CNN, 3D CNN backbones. The proposed TCA and TSA modules are also shown to be helpful. But the evaluation part should be improved:
1. Matrix Power Normalization is a n existing technique, therefore, in Table.1, the "+Matrix Power Normalization" is better moved to the upper part of this table and it can viewed as a stronger baseline compare to Plain GCP.  The Major contribution of this paper is TCA, TSA and TC-COV, which bring gain of 1.6% in total.
2. The hyper-parameter k should be studied.
3. Plain GCP + Matrix Power Normalization, which is a straightforward extension from existing state-of-the-art image covariance pooling, should better be compared in Table 4(a) to make the gain of the proposed TCP more clear.  The baseline of GAP in table 4(a) is kind of weak.

Overall, I think this is a nice work and I am willing to raise the score if my concerns can be addressed in the rebuttal phase.

**Time Spent Reviewing:**

48

---

> ### Author Response · Authors · 2021-08-10
> **Title: Response to Comments of Reviewer zp2B**
>
>   We sincerely thank the reviewer for spending precious time on giving insightful and constructive comments, and greatly appreciate recognition of the novelty of our manuscript, including temporal-based spatial and channel attention mechanisms as well as covariance pooling mechanism of intra-frame and inter-frame covariances. We also thank the reviewer for positive comments on reasonable motivation, extensive experiments, good organization and presentation. In the following, we will provide point-by-point responses, and hope our answers could address all concerns of the reviewer.
>
> 1. Clarity: Inter-frame and intra-frame are misleading
>
>   Sorry for the typos. The terms of inter-frame and intra-frame should be swapped, and we will modify them in the revised manuscript.
>
> 2. Clarity: Discussion and comparisons with ActionVLAD, NeXtVLAD and AttentionClusters.
>
>   Thanks for pointing out the related works. Going beyond VLAD for image classification, ActionVLAD, NeXtVLAD and AttentionCluster aim to aggregate local spatio-temporal features into a compact video representation based on multiple clusters. Specifically, ActionVLAD is an end-to-end local feature aggregation method, which summarizes residual vectors of spatio-temporal features across entire video relative to learnable clusters in a soft manner. NeXtVLAD can be regarded as a variant of VLAD, where encoding of high-dimensional features is decomposed to grouped encoding of multiple low-dimensional features, aiming to improve efficiency of VLAD. AttentionClusters is the backbone of the winner solution at ActivityNet Kinetics Challenge 2017, which splits features of an entire video into multi-granularity (multi-level) ones along channel and temporal dimension, while introducing a self-attention mechanism to perform interaction among multi-granularity features. As such, AttentionCluster can obtain an effective coarse-to-fine video representation. Different from them, TCP computes both intra-frame and inter-frame covariances of calibrated spatio-temporal features to produce powerful video representations.
>
>   As suggested by the reviewer, we compare with ActionVLAD, NeXtVLAD and AttentionClusters on four benchmarks, including K-400, UCF-101, HMDB-51 and S-S v1. The comparison results are summarized in the table below, in which the results are obtained using RGB information only, unless otherwise specified. The results of ActionVLAD, NeXtVLAD and AttentionClusters are duplicated from the paper of AttentionClusters. As seen in the table below, our TCP performs much better than ActionVLAD on all four datasets.  Besides, our TCP outperforms AttentionClusters on S-S V1  and HMDB-51.  On Kinetics-400, TCP with backbone of R-50 performs on par with AttentionClusters with backbone of Inception-ResNet-v2 (Inc-Res), while on UCF-101 our TCP with R-50 and RGB input is slightly better than AttentionClusters with R-152 that uses modalities of RGB and Flow. We mention that, on Kinetics-400,  AttentionClusters further achieves significant improvement by integrating RGB, Flow and Audio modalities. Additionally, our TCP achieves 42.6% mAP on Charades, while ActionVLAD obtains 21.0% mAP. On Mini-K200, our TCP performs better than ActionVLAD over 3.1% using backbone of 2D R-50. We will add discussion and results above in the revision.
>
> | | Kinetics-400 | S-S V1 (TSN) |S-S V1 (TSM) |    UCF-101 |    HMDB-51 |
> |:-----|:------:|:------:|:--------:|:------:|:------:|
> |ActionVLAD |74.4 (Inc-Res) | 23.8 (Inc-Res)|43.0 (Incp-Res) |94.2 (R-152, RGB+Flow) | 68.7 (R-152, RGB+Flow) |
> |NeXtVLAD |N/A|23.0 (Inc-Res)|41.9 (Inc-Res) |N/A|N/A|
> |AttentionCluster |75.3 (Inc-Res) | 27.7 (Inc-Res)|43.7 (Inc-Res)  |94.9 (R-152, RGB+Flow) | 69.8 (R-152, RGB+Flow) |
> |TCP| **75.3** (R-50)| **38.3** (R-50)|**46.9** (R50) |**95.1** (R-50, RGB)| **72.5** (R-50, RGB) |
>
> 3. Significance: +MPN can be viewed as a stronger baseline
>
>   Thanks for your suggestion, and we will move +MPN to the upper part of Table 1. We would like to explain that our work introduces MPN to exploit geometry of temporal covariance representations, which, to our best knowledge, has not been studied previously for video tasks.  As presented in Table 2, simple usage of frame-wise iSQRT-A (widely used in image classification) brings no improvement over baseline, suggesting that successful application of MPN for video is not straightforward.  In fact, we perform a comprehensive study of global covariance pooling for deep video recognition architectures, and explore several baseline methods, establishing a strong baseline (i.e., covariance pooling across the while video with MPN). Starting from the strong baseline established by us, our TCP further obtains a non-trivial performance gain (1.6%).
>
> 4. Significance: The hyper-parameter $k$ should be studied.
>
>   Kindly mention that we have evaluated effect of the hyper-parameter $k$ in Table A1 and A2 of the supplementary material (due to space limitation). To facilitate reference of the reviewer, both tables are also attached below.  Particularly, we can draw the following conclusions from Table A1 and A2. (1) The best results are achieved when kernel size $k$ is approximately equal to half of frames number, i.e., $k=5$ for 8 input frames and $k=9$ for 16 input frames. (2) Small range of interaction (due to small $k$) is better than those without interaction (i.e., $k=1$), but inferior to large range of interaction. (3) When kernel sizes are larger than half of frames number, the performance consistently decreases and drops by about 1\% when all frame interactions are modeled, indicating temporal aggregation via interaction among all input frames is not favorable. In the revision, we will try to move them into main body of paper.
>
> |  $k$ (8 frames) | 1 | 3 | 5 | 7 |
> |:-----|:------:|:--------:|:------:|:------:|
> |Top-1 Acc(%) | 75.6| 76.0 | 76.1 |76.0 |
> |Top-5 Acc(%) | 92.5 | 92.7 | 92.7| 92.6 |
>
> | $k$ (16 frames ) | 1 | 3 | 5 | 7 | 9 | 11 | 13| 15 |
> |:-----|:------:|:--------:|:------:|:---:|:---:|:---:|:---:|:---:|
> |Top-1 Acc(%) | 77.0| 77.0 | 77.4 | 77.5|77.7 |77.3 | 76.9| 76.8 |
> |Top-5 Acc(%) | 93.0 | 93.1 | 93.3 | 93.3| 93.5 | 93.1 | 93.0 | 93.0 |
>
> 5. Significance: Results of Plain GCP + matrix power normalization in Table 4 (a)
>
>   As suggested by the reviewer, we conduct experiments to compare with Plain GCP + matrix power normalization (MPN) using K-400 in Table 4 (a), and results are shown in the table below. It can be seen that our TCP is superior to Plain GCP + matrix power normalization by 2.2%, 1.1%, 0.5% and 1.0% with deep video architectures of  TSN + R50, TSN + R152, X3D-M and TEA + R50, respectively. Note that Plain GCP + matrix power normalization is a strong baseline established by us, upon which our  TCP still can achieve non-trivial performance gains, demonstrating the effectiveness of our TCP in capturing complex temporal dynamics.
>
> |Backbone | GAP | Plain GCP + MPN | TCP |
> |:-----|:------:|:--------:|:------------:|
> |TSN + R50 |70.6| 73.1 | **75.3** |
> |TSN + R152| 75.9 | 77.2 | **78.3** |
> |X3D-M | 76.0  | 76.5  |   **77.0**|
> |TEA + R50 |75.0| 75.8 | **76.8** |

---

> > ### Comment · Reviewer_zp2B · 2021-08-21
> > **Response to author rebuttal**
> >
> > Most of my problems are addressed in the authors' response. Though other reviewers have concerns on novelty of this paper, I still think this work is not a naive extension of plain covariance pooling, the proposed temporal attentive covariance pooling is not simply combination of self-attention and covariance pooling. So I still recommend an accept for this paper.

---

> > > ### Author Response · Authors · 2021-08-21
> > > **Response to comment of Reviewer zp2B**
> > >
> > >   We are pleased to know that our response addressed most of the reviewer's concerns. Once again, we sincerely thank the reviewer for recognition of our work's novelty and recommending accept of our paper.

---

### Official Review · Reviewer_SHTG · 2021-07-16

**Rating:** 6
**Confidence:** 4

**Summary:**

This paper presents a temporal-attentive covariance pooling (TCP) method for action recognition, which consists of a temporal attention module, a temporal covariance pooling, and a fast matrix power normalization of covariance. TCPNet is then devised by inserting TCP at the top of existing deep architectures. The experiments are conducted on 6 action recognition benchmarks to evaluate the proposed model.

**Limitations And Societal Impact:**

I do not find any significant aspect about its limitation and potential negative societal impact.

**Main Review:**

Strengths:

++ The paper is clearly written and well organized.

++ Extensive experiments are conducted on 6 benchmarks.

Weaknesses:

-- The technical contribution of this work is somewhat limited. The main idea of temporal-attentive covariance pooling seems to be a simple combination of self-attention mechanism and deep covariance pooling which have been widely used for action recognition [7, 17]. The combination of these techniques could be a contribution. But to me, the technical contribution is somewhat incremental for a NeurIPS paper.

-- In Table 4(a), the authors only compared the proposed TCP to one simple pooling method (GAP). It is better to include more feature aggregation baselines (such as ActionVLAD[18] and TLE[7]) for performance comparison.

-- In section 4, the authors should compare the proposed TCPNet with the state-of-the-art methods, such as [a] (81.0% on Kinetics-400, 53.3% on S-S V1) and [b] (50.6% on S-S V1).

[a] Heng Wang, Du Tran, Lorenzo Torresani, and Matt Feiszli. "Video modeling with correlation networks." In CVPR, 2020.

[b] Swathikiran Sudhakaran, Sergio Escalera, and Oswald Lanz. "Gate-shift networks for video action recognition." In CVPR, 2020.

-- It is better to show some qualitative results of the learnt temporal attention.


**Time Spent Reviewing:**

8 hours

---

> ### Author Response · Authors · 2021-08-10
> **Response to Comments of Reviewer SHTG**
>
>   Thanks for the positive comments on our clear writing, good organization and extensive experiments. In the following, we will carefully answer all questions, and sincerely hope our response could address all concerns of the reviewer.
>
> 1. A simple combination of self-attention mechanism and deep covariance pooling which have been widely used for action recognition [7,17].
>
>   We would like to clarify that **our temporal-attentive covariance pooling (TCP) is clearly different from previous works [7,17]**, and main differences are summarized as follows. (1) Our TCP makes the first attempt to learn second-order, intra-frame and inter-frame relations of spatio-temporal features. In contrast, TLE [7] directly uses B-CNN [26] or CBP [15] to compute covariances of temporal aggregated features, while AP [17] formulates bottom-up and top-down attention as a low-rank approximation of bilinear pooling (B-CNN [26]). AP can be implemented by an element-wise multiplication on features outputted by two parallel branches. (2) Both TLE [7] based on B-CNN/CBP and AP [17] based on low-rank approximations of bilinear pooling consider no geometry of covariance spaces, while our TCP makes it clear that matrix power normalization (MPN) exploiting geometry of covariance spaces is important for video application. (3) For use of temporal information, our TCP computes intra-frame and inter-frame covariances of temporal-attentive features by performing interaction across different frames. Differently, TLE [7] computes covariances of features aggregated by performing dot product along temporal dimension, while AP [17] focuses on computing frame-wise attention without explicitly temporal modeling. As a result of the big differences highlighted above, our TCP is significantly better than TLE [7], as shown in Table 2 (BCNN-A and CBP-A) and Table 7. Compared with AP using only RGB information, our TCP obtains accuracies of over 70% on HMDB51, while AP only achieves 52.2%.
>
>   The attention mechanisms of AP [17] and ours differ greatly. Our method is concerned with **temporal-aware spatial as well as channel attention**, while AP [17] essentially consists in a low-rank approximation of bilinear pooling, which focuses on **spatial attention of single frame**, considering no temporal information. **Furthermore, our temporal-based attention is also clearly different from other existing self-attention mechanisms, e.g., Non-local [57] or SE in X3D [10].** Specifically, our temporal-based spatial attention module (TSA) considers spatial correlation among three consecutive frames along temporal dimension while Non-local considers the correlation among all spatio-temporal features of whole video clip in a both spatial- and temporal-orderless manner. Our TSA is not only more efficient than Non-local (i.e., 0.16 GFLOPs vs. 0.7 GFLOPs), but also achieves better performance--accuracy of our TCP decreases from 76.1% to 75.6%, if our spatial attention module TSA is replaced by the Non-local module. Our temporal-based channel attention module (TCA) considers the temporal difference between consecutive frames, while SE for X3D computes attention only depending on individual single frames.  When our TCA is replaced by the naïve SE, accuracy of our TCP decreases from 76.1% to 75.6%, indicating that the proposed TCA is more effective.
>
>   Additionally, we would like to mention that, from technical perspective, it is not straightforward to introduce covariance pooling for boosting performance of the state-of-the-art video recognition methods. As shown in Table 1 and Table 2, Plain GCP and previous TLE method [7] (i.e., BCNN-A and CBP-A, see Table 2) bring little performance gains over GAP, while simple frame-level iSQRT-A (widely used in image classification) also brings no improvement. Our work is among the first to perform a comprehensive study of global covariance pooling for deep video recognition architectures, and significantly improves various popular deep video recognition architectures on six widely used benchmarks.
>
>   In summary, our TCP aims to effectively capture complex temporal dynamics, which is accomplished by learning both intra-frame and inter-frame correlations of the calibrated (attentive) spatio-temporal features in a unified manner. Besides, both our temporal-aware attentions and global covariance pooling are clearly different from existing methods, e.g., [7] and [17] pointed by the reviewer. We argue that, technically, our TCP is not a simple combination of widely used self-attention mechanism and deep covariance pooling but a novel unified spatio-temporal covariance pooling module.  Practically, our work provides an effective and affordable solution to introduce covariance pooling for boosting performance of a variety of popular 2D, 3D and pseudo-3D CNN architectures. We sincerely hope the reviewer could reconsider overall evaluation on the technical contributions of our work.
>
> 2. Comparison with TLE and ActionVLAD
>
>   Kindly note that we have compared with TLE [7] in Table 2 and Table 7 on Mini-K200, UCF-101 and HMDB-51 datasets. Specifically, TCP outperforms TLE (i.e., BCNN-A [7] and CBP-A [7] in Table 2) by about 4% on Mini-K200, while performing on par or better on UCF-101 and HMDB-51, though TLE employs both RGB and optical flow while our TCP only uses RGB information. Following the reviewer's suggestion, we give the comparison with ActionVLAD. On Mini-K200, our TCP is better than ActionVLAD by 3.1% in top-1 accuracy. On UCF-101 and HMDB-51, TCP respectively obtains 1.5% and 2.7% gains over ActionVLAD, although ActionVLAD combines CNN-based VLAD representation and extra iDT-based representation.   Additionally, our TCP clearly outperforms ActionVLAD on K-400 and S-S V1, despite that ActionVLAD uses a stronger backbone (i.e., Inc-Res) than our TCP (i.e., R-50). To facilitate comparison, the result are summarized in the table below, and we will add results of ActionVLAD in the revision.
>
>
> |Method| K-400 |S-S V1 (TSN)|S-S V1 (TSM) |UCF-101 | HMDB-51|Mini-K200 |
> |:----|:---|:---:|:---:|:--------:|:------------:|:------------:|
> |ActionVLAD [18] | 74.4 (Inc-Res)|  23.8 (Inc-Res)|43.0 (Inc-Res) | 93.6 (+iDT) | 69.8 (+iDT)|73.0 (R-50)|
> |TLE [7] | N/A | N/A | N/A |95.6 (+Flow) | 71.1 (+Flow) |72.2 (R-50) |
> |TCP (Ours)| **75.3** (R-50) |**38.3** (R-50)|**46.9** (R-50) | **95.1** (TSN)/**96.4** (TEA) | **72.5** (TSN)/**76.7** (TEA) |**76.1** (R-50)  |
>
>
> 3. Comparison with CorrNet [a] and GSM [b]
>
>   Thanks for the suggestion, and we will compare with CorrNet [a] and GSM [b] in the revision. Specifically, CorrNet [a] learns motion patterns by computing similarities of each feature in one frame with its spatially neighboring features in the adjacent frame.  GSM [b] performs gated temporal and spatial transformations along temporal dimension in a competitive way, which is embedded into 2D CNN for video recognition. On K-400, CorrNet achieves 81.0% using backbone of 3D R-101 pretrained on large-scale action dataset of YouTube8M. Our TCP based on X3D-M (77.0%) performs on par with CorrNet with backbone of 3D R-50 (77.2%). For 8 input frames, GSM achieves 49.0% in Top-1 accuracy on S-S v1, which is inferior to our TCP (49.6%). GSM achieves 50.6% in Top-1 accuracy when 16 input frames are used.
>
> 4. Qualitative results of the learnt temporal attention
>
>   Thanks for the suggestion. We visualize results of the learnt temporal attention using some example videos on K-400. As the system does not allow to typeset figure, we illustrate the visualization results at an anonymous URL https://github.com/erwrfdsavcz/Visualization. The qualitative results show the attention module in our TCP can effectively focus on key moving parts (e.g., hands and food in “tasting food”) for action recognition, while suppressing the remaining irrelevant regions.

---

> > ### Comment · Reviewer_SHTG · 2021-08-26
> > **post-rebuttal**
> >
> > I appreciate the efforts made by the authors to clarify the differences between TCP and existing works. Moreover, the authors addressed the issue of comparisons with more state-of-the-art methods. Therefore, I tend to upgrade my rating to "6: Marginally above the acceptance threshold".

---

> > > ### Author Response · Authors · 2021-08-27
> > > **Response to comment of Reviewer SHTG**
> > >
> > > We are delighted to know that our responses addressed the reviewer's concern about the differences between TCP and existing works, as well as the concern about comparisons with more state-of-the-art methods. We sincerely thank the reviewer for acknowledgement of our contributions and upgrading the overall rating to "6: Marginally above the acceptance threshold".

---

### Official Review · Reviewer_qjEJ · 2021-07-18

**Rating:** 6
**Confidence:** 4

**Summary:**

The paper proposes a second-order pooling method, called temporal-attentive covariance pooling (TCP), for summarizing temporal features and generating video representation. Specifically, TCP involves three components: 1) a temporal convolution before covariance pooling that introduces learnable parameters for temporal modeling; 2) a spatial and channel attention to further enhance the convolutional features; 3) matrix power normalization to utilize geometry of covariance space. Experiments on multiple video datasets show that TCP outperforms the standard global average pooling and simple covariance pooling.


After author rebuttal:
The author responses address most of my major concerns: (1) Contribution/novelty: after carefully reviewing the paper, the review from other reviewers and the author's rebuttals, I agree to recognize the contribution of TCP for introducing a solid framework that successfully applies the second-order pooling method to boost video action recognition. (2) TSA design: although the method is not intuitive to me, the comparison results provided in the author rebuttal justify the effectiveness of the proposed design.
In all, I'll raise my rating to "6: Marginally above the acceptance threshold".

**Limitations And Societal Impact:**

The paper didn't discuss its limitations and potential negative societal impact.

**Main Review:**

My major concern of this paper is its limited technical contribution -- most components of TCP are not novel and have been widely studied in prior work.

1) TCP aims to improve the vanilla covariance pooling by introducing order-aware aggregation along temporal dimension. However, simply adding a temporal convolution before covariance pooling is not sufficient for achieving this goal. First, temporal convolution can only capture local temporal dependencies within a small sliding window. The video-level temporal aggregation is still unachievable because the covariance features are still pooled by simple average operation. Moreover, using temporal convolution to capture local temporal dependencies is not novel and has been widely used for video recognition.

2) The proposed temporal-based spatial attention and channel attention are not novel as well. Both spatial attention [1] and channel attention [2] have been widely used for video recognition, and their combination is also presented in prior work, for example [3]. Moreover, the motivation of using $X_{l-2}, X_{l-1}$ as query and key for $X_l$ is not clear. $X_{l-2}$ may have very different data distribution with $X_{l}$, which makes the alignment of the generated attention map and the feature map $X_{l}$ inaccurate.

3) According to Table 1, the contribution of matrix power normalization is the most significant in TCP. However, matrix power normalization is proposed and used for vision tasks in prior work and it is simply applied to TCP in this paper.

Minor issue: The terms "intra-frame" and "inter-frame" in Section 3.2 is misleading or even incorrect. Correlation within the same frame $X_l^T X_l$ should be intra-frame covariance instead of inter-frame covariance, and vice versa.

In all, although the proposed TCP achieves superior accuracy compared with simple average pooling and covariance pooling, the novelty of TCP is limited and more careful designs to improve the temporal aggregation of covariance features are needed. In addition, considering the extra computation and parameter introduced in computing covariance pooling, spatial/channel attention and matrix power normalization, the performance gain of TCP is not that significant, especially for 3D networks that already consider local temporal dependencies.

[1] Wang, Xiaolong, et al. "Non-local neural networks."
[2] Feichtenhofer, Christoph. "X3d: Expanding architectures for efficient video recognition."
[3] Woo, Sanghyun, et al. "Cbam: Convolutional block attention module."


**Time Spent Reviewing:**

8

---

> ### Author Response · Authors · 2021-08-10
> **Response to Comments of Reviewer qjEJ**
>
> (1) Response to technical contribution/novelty of our paper
>
>   We thank the reviewer for spending valuable time on reviewing our paper. Kindly note that the reviewer seems not to be fully aware of the individual components of temporal-attentive covariance pooling (TCP), and also seems to neglect the contribution of the holistic architecture design leading to competitive video recognition performance. We sincerely hope our point-by-point responses could address the concerns and the reviewer could reconsider the recommendation.
>
>   First of all, we would like to clarify that the holistic architecture of the proposed second-order pooling is clearly different from existing methods. To our best knowledge, our TCP makes **the first attempt to characterize both intra-frame and inter-frame correlations of the attentive spatio-temporal features**, leading to strong ability to capture complex temporal dynamics. Besides, we would like to note that, from technical perspective, it is not straightforward to introduce covariance pooling for boosting performance of state-of-the-art architectures. As shown in Table 1 and Table 2, Plain GCP and previous TLE method [7] (i.e., BCNN-A and CBP-A) bring little performance gains over GAP, while frame-level iSQRT-A (widely used in image classification) also brings no improvement. In contrast, **our work is among the first to make a comprehensive study of global covariance pooling in the field of video recognition, and significantly improves various popular deep video recognition architectures on six widely used benchmarks**. In the following, we will carefully address the reviewer's concerns about individual components of our TCP and sincerely hope the reviewer could reconsider overall evaluation on technical contribution/novelty of our work.
>
> 1. Temporal aggregation via temporal convolution in TCP
>
>   Kindly note that TCP aims to **characterize both intra-frame and inter-frame correlations** among different frames as seen in Eqn. (5), which can be judiciously **implemented by temporal convolution** with kernel size of $1\times 1\times k$ in Eqn. (10). As such, the range of dependency among different frames is determined by kernel size $k$ of the temporal convolution. For example, if kernel size $k$ is equal to number of input frames, TCP will learn video-level temporal dependency, by modeling correlations of all input frames. The effect of kernel size $k$ was evaluated in Tables A1 and A2 of the supplementary material. To facilitate reference of the reviewer, both tables are also attached below, from which we can draw the following conclusions. (1) The best results are achieved when kernel size $k$ is approximately equal to half of frames number, i.e., $k=5$ for 8 input frames and $k=9$ for 16 input frames. (2) Small range of interaction (due to small $k$) is better than those without interaction (i.e., $k=1$), but inferior to large range of interaction. (3) When kernel sizes are larger than half of frames number, the performance consistently decreases and drops by about 1% when all frame interactions are modeled, **indicating temporal aggregation via interaction among all input frames is not favorable**. In our work, kernel size of temporal convolution in TCP is set to about half of frames number, and we underline its purpose is to capture the most suitable range of second-order, cross-correlations of different frames.
>
>
> |  $k$ (8 frames) | 1 | 3 | 5 | 7 |
> |:-----|:------:|:--------:|:------:|:------:|
> |Top-1 Acc(%) | 75.6| 76.0 | 76.1 |76.0 |
> |Top-5 Acc(%) | 92.5 | 92.7 | 92.7| 92.6 |
>
> | $k$ (16 frames ) | 1 | 3 | 5 | 7 | 9 | 11 | 13| 15 |
> |:-----|:------:|:--------:|:------:|:---:|:---:|:---:|:---:|:---:|
> |Top-1 Acc(%) | 77.0| 77.0 | 77.4 | 77.5|77.7 |77.3 | 76.9| 76.8 |
> |Top-5 Acc(%) | 93.0 | 93.1 | 93.3 | 93.3| 93.5 | 93.1 | 93.0 | 93.0 |
>
> 2. Response to novelty of temporal-based spatial and channel attentions
>
>   Above all, we mention that our temporal-based spatial and channel attention have clear motivation to model complex temporal dynamics, very different from existing methods, e.g., Non-Local network [1] and SE for X3D [2] pointed out by the reviewer. Specifically, **our temporal-based spatial attention module (TSA) considers spatial correlation among three consecutive frames along temporal dimension**, while Non-local considers the correlation among all spatio-temporal features of whole video clip in a spatial- and temporal-orderless manner. Our TSA is not only more efficient than Non-local (i.e., 0.16 GFLOPs vs. 0.7 GFLOPs), but also achieves better performance--accuracy of our TCP decreases from 76.1% to 75.6%, if our spatial attention module TSA is replaced by the Non-local module.  **Our temporal-based channel attention module (TCA)** considers the temporal difference between consecutive frames, while SE for X3D computes attention only depending on individual single frames.  When our TCA is replaced by the simple SE, accuracy of our TCP decreases from 76.1% to 75.6%, indicating the proposed TCA is more effective. Additionally, our method is clearly different from CBAM, which models spatial and channel attentions for image without considering temporal information.
>
>   We argue that the motivation of the proposed TSA is clear in that we intend to model spatial attention by exploiting temporal information. Although objects may move across frames, the spatial movement is not very drastic between consecutive frame in most cases. Our TSA is implemented by adopting $X_{l}$ as values and $X_{l-2}$ and $X_{l-1}$ as queries and keys, respectively. To evaluate effect of possible misalignment, we compute a variant of TSA where only two consecutive frames are considered, i.e., $X_{l-1}$ as queries and $X_{l}$ as both keys and values, which yields 75.5% in top-1 accuracy, lower than TSA (76.1%) exploiting three consecutive frames. Besides, we also visualize our temporal-based attention using some sample video clips on Kinetics-400. The figures for visualization can be found at an anonymous URL https://github.com/erwrfdsavcz/Visualization, where our temporal-based attentions can effectively focus on the key parts of individual actions. Both qualitative and quantitative results above show our TSA is effective and reasonable.
>
> 3. Response to contribution of matrix power normalization
>
> Kindly note that we did not claim matrix power normalization (MPN) is our contribution; instead, we clarify that our contribution lies in that how to make covariance pooling with MPN perform competitively by careful and extensive study in the video recognition task, which, to our best knowledge, has not been studied previously. As listed in Table 2, although MPN is used, frame-level iSQRT-A (widely used in image classification) also brings no improvement over baseline GAP. Therefore, how to exploit MPN for covariance pooling in video is not straightforward. In this work, we make the first attempt to perform a comprehensive study of global covariance pooling for deep video recognition architectures, and established several baseline methods, including a strong baseline using matrix power normalization (i.e., video-level iSQRT). We argue that this strong baseline is our contribution, and upon it, our TCP further obtains non-trivial performance gains (1.6%).
>
> (2) Minor issues:  Inter-frame and intra-frame are misleading.
>
> Sorry for the typos. The terms of inter-frame and intra-frame should be swapped, and we will modify them in the revised version.
>
> (3) Extra computation vs. performance improvement & gains over 3D backbone
>
> As stated in Sec. 3.3, our TCP brings about extra 13.6% (5.6%) parameters and 3.6% (1.3%) FLOPs for 2D R-50 (2D R-152). However, TCP brings 4.7% and 2.4% performance gains for R-50 and for R-152 respectively, and we believe the improvement is significant in terms of accuracy and computation. For 3D backbone, our TCP brings 1.0% improvement over X3D-M. Kindly note that X3D-M is a very strong baseline, obtained by extensive architecture search along network axes of space, time, width and depth on K-400 dataset. We believe 1.0% gains over such an architecture is non-trivial. For reference, we would like to mention two recent works [R1] and [R2]-- the former achieves 0.6% gains over X3D-M, while the latter achieves 0.6% and 0.2% gains over X3D-S and X3D-XL, respectively.
>
> [R1] A. Stergiou, R. Poppe. Multi-Temporal Convolutions for Human Action Recognition in Videos. In IJCNN, 2021.
>
> [R2] M. Fayyaz, E. Bahrami, A. Diba, et al. 3D CNNs with Adaptive Temporal Feature Resolutions. In CVPR, 2021.

---

> > ### Comment · Reviewer_qjEJ · 2021-08-23
> > **Response to Author Comments**
> >
> > I'd like to thank the authors for their detailed responses to my questions and concerns. The author responses address most of my major concerns: (1) Contribution/novelty: after carefully reviewing the paper, the review from other reviewers and the author's rebuttals, I agree to recognize the contribution of TCP for introducing a solid framework that successfully applies the second-order pooling method to boost video action recognition. (2) TSA design: although the method is not intuitive to me, the comparison results provided in the author rebuttal justify the effectiveness of the proposed design.
> >
> > In all, I'll raise my rating to "6: Marginally above the acceptance threshold".

---

> > > ### Author Response · Authors · 2021-08-24
> > > **Response to comment of Reviewer qjEJ**
> > >
> > > We sincerely thank the reviewer for carefully reviewing our paper and attentively reading our responses. We are delighted to hear that our responses addressed most of the reviewer's major concerns, especially about contribution/novelty of TCP and TSA design. We will add the qualitative and quantitative analyses on TSA in the revised version. Finally, we greatly appreciate the reviewer's recognition of our work and the decision on raising the overall rating to "6: Marginally above the acceptance threshold".

---

### Decision · Program_Chairs · 2021-09-27

**Decision:**

Accept (Poster)

**Comment:**

All three reviewers provided a rating of "6: Marginally above the acceptance threshold" for this submission. Reviewers qjEJ and SHTG initially raised concerns about the technical novelty of the proposed temporal-attentive covariance pooling (TCP). However, the authors provided a response that convinced both reviewers that there are significant differences between the TCP and related mechanisms introduced in prior works. Reviewer qjEJ pointed out the somewhat unintuitive design of TSA but concurs that the approach is empirically effective. Finally, Reviewer zp2B requested a few additional studies (such the comparison to local feature aggregation methods and the ablation on k), which were presented in the response by the authors. The ACs agree with the recommendation of accepting the paper.